# Evolution of the Morphology and Magnetic Properties of Flaky FeSiAl/MFe_2_O_4_ (M = Mn, Co, Ni, Cu, Zn) Composites

**DOI:** 10.3390/nano13040712

**Published:** 2023-02-13

**Authors:** Chuannan Ge, Chenglong Lei, Bo Wang, Yakun Wang, Zhouhao Peng, Zhitong Wang, Yunjun Guo

**Affiliations:** 1School of Physics and Information Engineering, Jiangsu Second Normal University, Nanjing 210013, China; 2Haian Institute of High-Tech Research, Nanjing University, Nanjing 210093, China

**Keywords:** nanosheet, flaky FeSiAl, spinel ferrite, magnetism, co-precipitation

## Abstract

Nanosized spinel ferrites MFe_2_O_4_ (M = Mn, Co, Ni, Cu, Zn)-coated flaky FeSiAl alloy composites were synthesized successfully. Nano-ferrites preferentially grow into nanoplatelets due to induced or restricted growth on the flaky surface of FeSiAl. With annealing temperature increasing, the ferrites’ nanosheets thicken gradually and then grow into irregular particles. The annealing temperature not only affects the nanosized morphology and coating but also the magnetic properties of flaky FeSiAl composites. The saturation magnetization of CuFe_2_O_4_- or NiFe_2_O_4_-coated FeSiAl is approximate 69 emu/g, where the value of MnFe_2_O_4_-, CoFe_2_O_4_- and ZnFe_2_O_4_-coated FeSiAl show a decreasing trend generally from 64 emu/g to 55.7 emu/g annealing at 800 °C, respectively. The saturation magnetization of flaky FeSiAl composites was improved with the increased annealing temperature, except for those coated with ZnFe_2_O_4_ and NiFe_2_O_4_. These results are useful for improving the comprehensive properties of ferrite-coated flaky FeSiAl alloy composites.

## 1. Introduction

Currently, Fe-based alloy composites have attracted considerable interest for magnetic powder cores, microwave absorption, and flexible noise suppression sheets due to their excellent magnetic properties [1,2,3,4,5]. The composite of ferrite materials on the surface of an FeSiAl alloy has been an effective method used to regulate magnetic properties. For example, to solve problems of low permeability and high hysteresis loss for the powder core, one effective solution is using the magnetic material with a high resistivity to form the insulating layer [3,6,7]. In addition, the practical application of the pure flaky FeSiAl for excellent microwave absorption is still limited by magneto-electric properties. Thus, the most promising approach to solve the problem is constructing composites with other coating materials [8,9].

As we know, the construction of heterogeneous structures on the surface of an FeSiAl alloy not only forms a variety of rich micro-nano interfaces but also has a great impact on magnetic properties [10,11,12]. In terms of the construction method, various preparation routes, including sol–gel pyrolysis, co-precipitation and mechanical milling or mixing, have been used. Deng et al. [13] reported a facile chemical co-precipitation employed for fabricating the Fe_3_O_4_-modified flake-shaped FeSiAl. The modified sample shows marked decrease of the permittivity and slight reduction of the permeability. Zhou et al. [14] have prepared the spinel MnZn ferrites by solid-state reaction via mechanochemical method and mixed with FeSiAl powders. The result confirmed that by increasing ferrites content, the magnetization decreased. Jin et al. [15] obtained a flexible, soft magnetic composite by combining the FeSiAl flakes with varying fractions of NiZn ferrite particles via tape-casting method. They proposed that the composite filled with 14 wt% of ferrite particles showed the best coupling ratio at 1 MHz–1 GHz. Yang et al. [16] prepared the flaky FeSiAl coated with magnetic insulator NiCuZn ferrite by a sol–gel route. The result clearly showed that the permittivity of the obtained flaky FeSiAl decreased remarkably, accompanied by a slight decrease in permeability. Previous research [10,17,18,19] has mostly focused on the influence of magnetic ferrite type, coating content and particle size on the magnetic properties of Fe-based ally materials by different preparation paths. Generally, high performance can be enhanced by heat treatment, which affects phases, nanostructures, microscopic stresses and magnetic properties [20,21,22]. Due to the effect of annealing temperature, magnetic properties are affected by the phase transformation and grain formation. Stress relief also plays an important role. However, it has rarely been studied in terms of effect of annealed temperature on the microstructure and magnetic properties of flaky FeSiAl composites coated with different surface coatings. In addition, it is of great significance to systematically explore the regulation law of magnetic properties for different spinel ferrites-coated FeSiAl alloys.

In this article, nanosized spinel ferrites MFe_2_O_4_ (M = Mn, Co, Ni, Cu, Zn)-coated flaky FeSiAl composites were successfully prepared by simple and efficient co-precipitation. The evolution of morphology structure depending on annealing temperatures was investigated. To improve the comprehensive properties of ferrite-coated flaky FeSiAl alloy composites, the correlation of annealing temperatures, coating structures and magnetic properties for different ferrite were explored. The magnetism of different ferrites coating FeSiAl composites was also analyzed.

## 2. Materials and Methods

Pure flake-shaped FeSiAl (85 wt% Fe-9.6 wt% Si-5.4 wt% Al) alloy powders with a mesh size of 200 were commercially purchased. As shown in Figure 1, the nanosized spinel ferrites MFe_2_O_4_ (M = Mn, Co, Ni, Cu and Zn) were achieved by chemical co-precipitation. The synthesis was carried out using commercially available reagents: ferric sulphate (Fe_2_(SO_4_)_3_, AR), manganese sulfate (MnSO_4_·H_2_O, AR), cobalt nitrate (Co(NO_3_)_2_·6H_2_O, AR), cupric nitrate (Cu(NO_2_)_2_·6H_2_O, AR), nickel nitrate (Ni(NO_3_)_2_·6H_2_O, AR), zinc nitrate (Zn(NO_3_)_2_·6H_2_O, AR) and sodium hydrate. The weight ratio of ferrite to FeSiAl was designed to be 5 wt.%. Finally, the product was annealed under argon for 2 h at different temperatures. The product was carried out with the same route and labeled as @ MFe_2_O_4_ (M = Mn, Co, Ni, Cu, Zn).

X-ray powder diffraction (XRD, Rigaku Ultima IV, Japan) was performed with Cu Ka radiation (λ = 0.1541 nm). The morphology was characterized by scanning electron microscopy (SEM, Zeiss-Supra 55, Germany). Room temperature hysteresis (M–H) loops were characterized by vibrating sample magnetometer (VSM, vsm-175, China).

## 3. Results

The morphologies and magnetic hysteresis loop of the pure FeSiAl is shown in Figure 2. Clearly, the FeSiAl samples exhibit the typical flake shapes and smooth surface. The particle size of the FeSiAl is about 60–100 μm, and the particle thickness of the FeSiAl is less than 2 μm. The room temperature hysteresis (M–H) loops shown in Figure 2b proved to have excellent soft magnetic properties. This is similar to the magnetic properties previously reported [21].

Figure 3 gives the XRD patterns of the flaky FeSiAl/MFe_2_O_4_ (Mn, Co, Ni, Cu and Zn) samples annealing at 800 °C. Spinel ferrites phase was observed in the bcc structure of the FeSiAl alloy with the DO_3_ super-lattice, corresponding to the PDF card JCPDS #45-1206. The characteristic peaks at 30.3°, 35.5° and 43°—the crystal planes (220), (311) and (400), respectively—were observed to confirm the formation of a spinel ferrite phase at 800°C. [23]. A lower annealing temperature will be unfavorable to ferrite phase formation, as confirmed by the XRD pattern of composites coated with CuFe_2_O_4_ ferrites annealing at different temperature shown in Appendix A. The relative strengths of ferrite coatings are all very weak, indicating the surface-grown ferrite nanostructure, which was confirmed by subsequent SEM results. One can observe a slight offset of diffraction peak (311) corresponding to the ferrite at 35.5° and (400) at 43°due to a different cation radius [24,25]. The gray dashed lines corresponding to the spinel phase indicate the characteristic peak offsets of different ferrites. In addition, the strengths of (400) for CuFe_2_O_4_ and NiFe_2_O_4_ coatings are as strong as other ferrites. A possible reason is that the crystallization is relatively high.

Figure 4 shows the SEM images of the composites annealing at 800 °C, where the surface morphology and particle size of ferrites are obtained. The nano-ferrites are in situ growths on the surface of alloy flakes. From Figure 4i,l, it is seen that the NiFe_2_O_4_ and CuFe_2_O_4_ samples consist of regular-shaped nanoparticles. In addition, the particle size of CuFe_2_O_4_ is nanoparticle and the surface is not dense enough, as shown in Figure 4k,l. However, it is noticed that the MnFe_2_O_4_, CoFe_2_O_4_ and ZnFe_2_O_4_ samples display nanoplatelets, where the thickest nanoplatelets are observed in ZnFe_2_O_4_. Therefore, it can be seen that nano-ferrites preferentially grow in nanoplatelets on the FeSiAl surface, which may be the induced or restricted growth of crystal surfaces.

To study the effect of heat treatment on the growth structure of nano-ferrites, Figure 5 shows the SEM images of the representative flaky FeSiAl/CoFe_2_O_4_ composites annealing at temperatures of 650 °C, 700 °C, 750 °C, 800 °C and 850 °C. The results show the thickness of nanosheets, which thicken gradually from 11 nm to 25 nm with an increase in annealing temperature from 650 °C to 750 °C. When the annealing temperature reaches 850 °C, the crystals grow from nanosheets to irregular particles. The FeSiAl/ferrites composites were prepared by hydroxide co-precipitation. Therefore, under low temperature annealing, the nanosheets of ferrites were grown on the induced or restricted surface of flaky FeSiAl. Combined with the results of SEM in Figure 4, an evolution can be inferred that the transformation temperature for NiFe_2_O_4_ and CuFe_2_O_4_ samples is not more than 800 °C. 

Figure 6 shows the magnetic measurements through vibrating sample magnetometer (VSM) at room temperature for flaky FeSiAl coated with MnFe_2_O_4_, CoFe_2_O4, NiFe_2_O_4_, CuFe_2_O_4_ and ZnFe_2_O_4_ composites annealing under different temperatures. In general, the magnetic moment unit of the ferrite phase is smaller than that of the FeSiAl alloy phase. The saturation magnetization of pure FeSiAl was around 119 emu/g, but that of ferrites was less than 80 emu/g. Consequently, with ferrite coating, the atomic magnetic moment per unit volume or mass of FeSiAl/ferrites will inevitably be reduced, resulting in the decrease of saturation magnetization on the whole. As shown in Figure 6a, the saturation magnetization of FeSiAl@CuFe_2_O_4_ and FeSiAl@NiFe_2_O_4_ are approximately 69 emu/g, where the value of FeSiAl@MnFe_2_O_4_, FeSiAl@CoFe_2_O_4_ and FeSiAl@ZnFe_2_O_4_ show a decreasing trend generally from 64 emu/g to 55.7 emu/g, respectively. Theoretically, the magnetism of pure ferrites may be caused by differences in spinel types (normal/ inverse), distribution of cations, crystallization and nanosizes [23,25,26]. The dominant factors of the magnetic properties of FeSiAl/ferrites composites may depend on the crystallization and spinel type in current research.

Figure 6b shows the magnetic loop at room temperature for MFe_2_O_4_ (M = Mn, Co, Ni, Cu, Zn)-coated flaky FeSiAl composites annealing at 850 °C. The saturation magnetization of FeSiAl@MFe_2_O_4_ (M = Mn, Co, Cu) composites shows a larger value than that of annealing at 800 °C, except for FeSiAl@ ZnFe_2_O_4_ and FeSiAl@ NiFe_2_O_4_, which show a slight decrease. With the increase of the annealing temperature, the crystallization was favorable, as confirmed by the SEM result_,_ and then the saturation magnetization of the composite was improved. 

## 4. Discussion

The transformation temperature of morphology for NiFe_2_O_4_ and CuFe_2_O_4_ samples is not more than 800 °C, but for CoFe_2_O_4_, it reaches 850 °C. Annealing at 800 °C, the saturation magnetization of FeSiAl@CuFe_2_O_4_ and FeSiAl@NiFe_2_O_4_ are approximately 69 emu/g, where the value of FeSiAl@MnFe_2_O_4_, FeSiAl@CoFe_2_O_4_ and FeSiAl@ZnFe_2_O_4_ show a decreasing trend generally from 61 emu/g to 55.7 emu/g, respectively. The saturation magnetization of FeSiAl@MFe_2_O_4_ (M = Mn, Co, Cu) composites was improved with the increase of the annealing temperature, except for FeSiAl@ ZnFe_2_O_4_ and FeSiAl@ NiFe_2_O_4_. In the case of pure ferrite, zinc ferrite has zero magnetic moment due to the lack of free electrons in the Zn^2+^ and Fe^3+^ cancelling the magnetic moment by the spin directions, but in fact, zinc ferrite nanoparticles have a very low magnetic moment due to the cation distribution between tetrahedral A site and octahedral B site [24]. Thus, we assumed that the high annealing temperature may lead to the cation redistribution, which results in a lower magnetic moment for ZnFe_2_O_4_ and NiFe_2_O_4_ ferrites. In theory, MnFe_2_O_4_ ferrites have a larger saturation magnetism because Mn^2+^ has 5 Bohr magneton. The cobalt ferrite and the nickel ferrite have a medium saturation magnetism because Co^2+^ have 3 Bohr magneton and Ni^2+^ have 2 Bohr magneton. Thus, we observed a decrease in the saturation magnetization of the FeSiAl@MFe_2_O_4_ corresponding M = Co, Ni and Zn ions. It should be noted that we have an abnormal result in the saturation magnetization of FeSiAl@CuFe_2_O_4_ shown in Figure 6c,d. At a low annealing temperature, the saturation magnetization of the composite is similar to that of the pure FeSiAl. As we all know, the Cu^2+^ have 1 Bohr magneton in the CuFe_2_O_4_, and their magnetic properties when compared are theoretically lower than that of other ferrites. This may be due to nano-crystallization, as expected, and lower phase formation temperature. The XRD and SEM results shown in Figure 3b and Figure 4m, respectively, also confirmed this. It is reasonable to believe that the annealing temperature is conducive to the growth and cation redistribution of CuFe_2_O_4_ ferrite on the surface of flaky FeSiAl alloy.

## 5. Conclusions

The nanosized spinel ferrites MFe_2_O_4_ (M = Mn, Co, Ni, Cu, Zn)-coated flaky FeSiAl composites were successfully prepared via co-precipitation. XRD results show the spinel phase formation at low temperature of 800 °C. The SEM characterization shows that nano-ferrites preferentially grow into nanoplatelets due to the induced or restricted growth on the flaky surface of FeSiAl. With the increase of annealing temperature, ferrites’ nanosheets thicken gradually and the crystal melts from nanosheet to irregular particles. The transformation temperature of the Co ferrites coating is above 800 °C, but the other ferrites are lower. The annealing temperature not only affects the morphology of nanosized ferrites but also the magnetic properties of flaky FeSiAl/MFe_2_O_4_ (M = Mn, Co, Ni, Cu, Zn) composites. In addition, different spinel ferrites-coated FeSiAl alloys can regulate the magnetic properties of composites. The saturation magnetization value of Mn, Co and Cu ferrites coatings show an increasing trend generally with increasing temperature, but Ni and Zn ferrites coatings show the opposite. The value of the Cu ferrite coating is highest with 90 emu/g at 850°C and shows a value close to the pure FeSiAl powder at low annealing temperature. The hypothesis is that the high annealing temperature leads to the new cation distribution. These results are useful for improving the comprehensive properties of ferrite-coated flaky FeSiAl alloy composites, and are expected to promote further research on the correlation of annealing temperatures, coating structures and magnetic properties for different ferrites.

## Figures and Tables

**Figure 1 nanomaterials-13-00712-f001:**
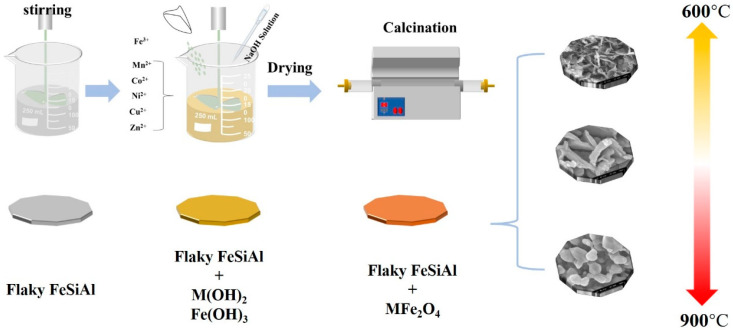
The schematic illustration of the evolution of MFe_2_O_4_ (M = Zn, Cu, Ni, Co and Mn)-coated flaky FeSiAl composites.

**Figure 2 nanomaterials-13-00712-f002:**
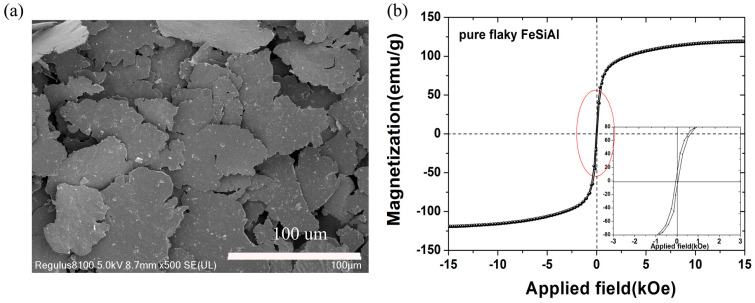
(**a**) SEM micrographs and (**b**) magnetic hysteresis loop of the pure FeSiAl.

**Figure 3 nanomaterials-13-00712-f003:**
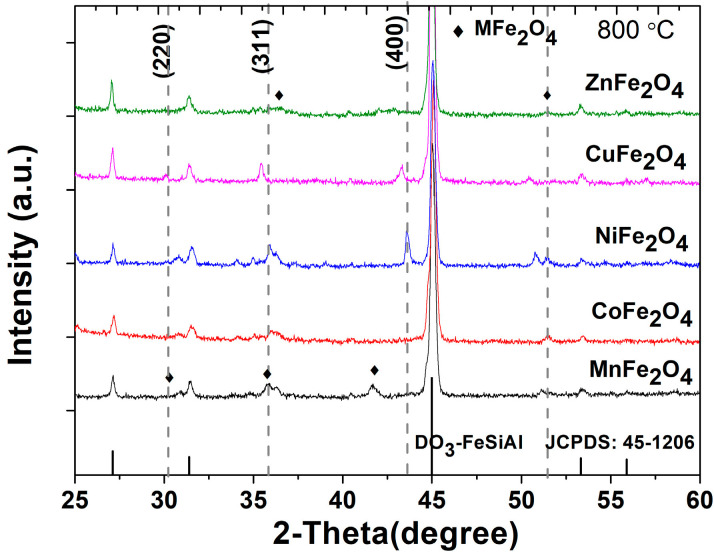
XRD pattern of flaky FeSiAl composites coated with MFe_2_O_4_ (M = Mn, Co, Ni, Cu, Zn) ferrites annealing at 800 °C.

**Figure 4 nanomaterials-13-00712-f004:**
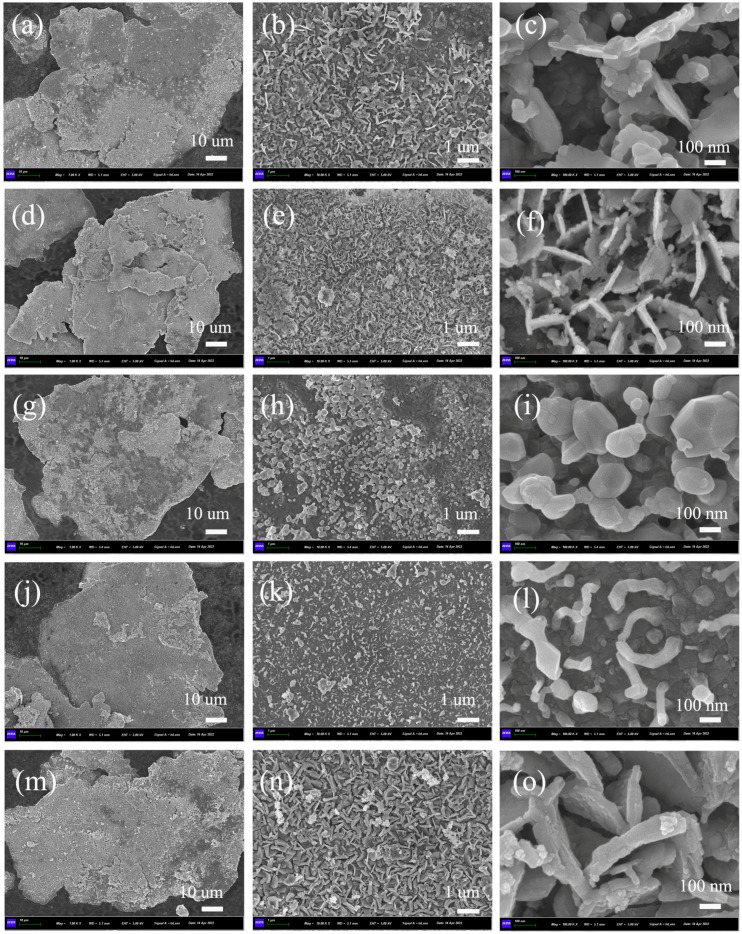
SEM of flaky FeSiAl composites coated with MFe_2_O_4_; M = (**a**–**c**) Mn, (**d**–**f**) Co, (**g**–**i**) Ni, (**j**–**l**) Cu, (**m**–**o**) Zn annealing at 800 °C.

**Figure 5 nanomaterials-13-00712-f005:**
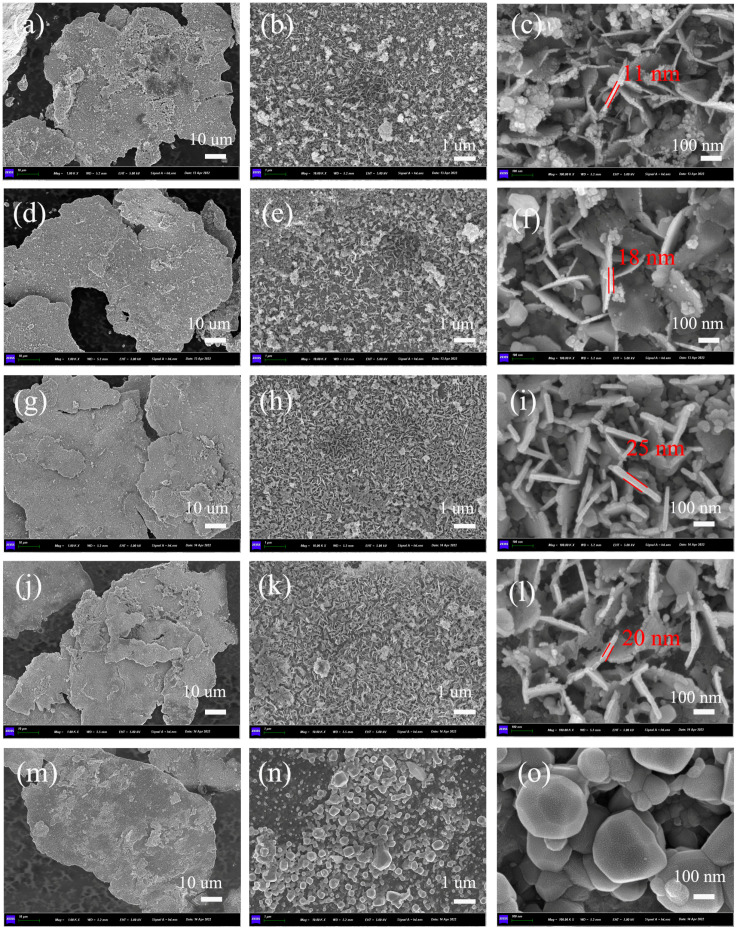
SEM of flaky FeSiAl composites coated with CoFe_2_O_4_ ferrites annealing at (**a**–**c**) 650 °C, (**d**–**f**) 700 °C, (**g**–**i**) 750 °C, (**j**–**l**) 800 °C, (**m**–**o**) 850 °C.

**Figure 6 nanomaterials-13-00712-f006:**
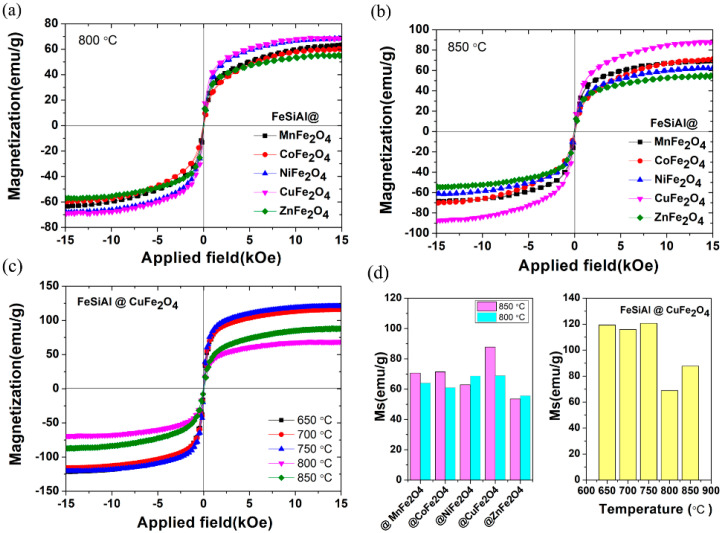
Hysteresis loops of spinel ferrite MFe_2_O_4_ (M = Mn, Co, Ni, Cu, Zn)-coated flaky FeSiAl composites annealing at (**a**) 800 °C and (**b**) 850 °C, (**c**) CuFe_2_O_4_ coated composites with different annealing temperature and (**d**) saturation magnetization of (**a**–**c**).

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
