# Peer review of "Evolution of the Morphology and Magnetic Properties of Flaky FeSiAl/MFe_2_O_4_ (M = Mn, Co, Ni, Cu, Zn) Composites"

_nanomaterials, 2023, doi:10.3390/nano13040712_

Round 1
Reviewer 1 Report
The article is interesting. It can be published after taking into account some remarks.please complete the abstract. It should contain the scope of the research and its originality.
The literature review is poor.Figure 2b is not very legible. Conclusions should be elaborate.
Author Response
Reviewer 1
The article is interesting. It can be published after taking into account some remarks.
1. please complete the abstract. It should contain the scope of the research and its originality.
Reply: As reviewer suggested, the abstract sections have been modified. The details are highlighted in red in the revised manuscript and the abstract contain the scope of the research and its originality.
- The literature review is poor.
Reply: Thanks for the mention of the reviewer. Considering the Reviewer’s suggestion, we do our best to improve the review.
- Figure 2b is not very legible.
Reply: As reviewer suggested, the Figure 2b was modified on page 3, line 86 in the revised manuscript.
- Conclusions should be elaborate.
Reply: Considering the Reviewer’s suggestion, the conclusion section was further elaborated.

Reviewer 2 Report
The article develops a successful co-precipitation procedure for coating flaky (commercial) FeSiAl particles with nanosized spinel ferrites.
Morphology evolution of the co-precipitated nanosheets with annealing temperature allows thickness evolution control to a final melt to irregular particles. These morphology changes are connected to magnetic properties evolution rendering an interesting approach for tailoring magnetic properties of the FeSiAl particles.
The verified conclusions open an interesting route for these interesting materials that should be explored through an analysis of competing reaction kinetics not explored on the article.
Author Response
Reviewer 2
The article develops a successful co-precipitation procedure for coating flaky (commercial) FeSiAl particles with nanosized spinel ferrites. Morphology evolution of the co-precipitated nanosheets with annealing temperature allows thickness evolution control to a final melt to irregular particles. These morphology changes are connected to magnetic properties evolution rendering an interesting approach for tailoring magnetic properties of the FeSiAl particles. The verified conclusions open an interesting route for these interesting materials that should be explored through an analysis of competing reaction kinetics not explored on the article.
Reply: Thanks for the mention of the reviewer. It is very correct to discuss the reaction kinetics, so as to understand the mechanism. However, due to the limitations of the research, we will further explore in the future work. It also provides reference for our future research direction. A suitable mechanism is qualitatively attempted in the manuscript, as the composite of two types of magnetic materials, especially the surface of sheet metal materials growing nano ferrites. Since the precipitating precursors are all hydroxides pyrolysis at low temperature to form spinel ferrites, the constrained growth of the interface may be the main reason for the formation of ferrite nanosheets, which is also explained by the further increase of annealing temperature to form random particles.

Reviewer 3 Report
In the submitted manuscript, the FeSiAl flakes were covered with various ferrites by co-precipitation method. The resultant composite was investigated by means of SEM, XRD and magnetic measurements. SEM showed that the flakes were indeed covered with some nanoparticles. XRD confirmed DO3-structure of the flakes not explicitly revealing the reflections from the ferrites. The magnetic measurements showed that after covering the flakes with nanoparticles, the hysteresis is still present with lower magnetization saturation values than without any coverage.
I recommend the major revision for this article.
MAJOR:
- In the Introduction section it’s written on the coverage of alloy flakes with various nanoparticles in order to expand absorption band in the microwave range and slightly affect magnetic properties, as far as I understood. The articles aim is to investigate magnetic properties. In my opinion, there is an inconsistency between the Introduction section and the article’s aim.
- XRD in the fig. 3 didn’t explicitly revealed ferrites structures as the intensities of the respective reflections are almost invisible. SEM data didn’t contain any diffraction from the nanoparticles. The authors need to provide the evidence that the precursors were completely decomposed with the ferrites formation.
MINOR:
- It’s unclear if magnetic properties of the FeSiAl flakes were affected after the heat-treatment. Please provide the respective measurements.
- No XRD data is provided for the heat-treatments at 650, 700, 750 and 850oC.
- TYPOS and STYLE:
- Page 1, line 30: «absorbent» should be «absorber»
- Fig. 3: description for (a) and (b) sub-figures are missing
- Fig. 4 and 5: obviously (a), (d), (h), (k) and (n) sub-figures are too similar with each other. One sub-figure is enough. Others can be transferred into a supplementary information file.
Author Response
Reviewer 3
In the submitted manuscript, the FeSiAl flakes were covered with various ferrites by co-precipitation method. The resultant composite was investigated by means of SEM, XRD and magnetic measurements. SEM showed that the flakes were indeed covered with some nanoparticles. XRD confirmed DO3-structure of the flakes not explicitly revealing the reflections from the ferrites. The magnetic measurements showed that after covering the flakes with nanoparticles, the hysteresis is still present with lower magnetization saturation values than without any coverage. I recommend the major revision for this article.
- In the Introduction section it’s written on the coverage of alloy flakes with various nanoparticles in order to expand absorption band in the microwave range and slightly affect magnetic properties, as far as I understood. The articles aim is to investigate magnetic properties. In my opinion, there is an inconsistency between the Introduction section and the article’s aim.
Reply: Thanks for the mention of the reviewer. The introduction section was to introduce important applications of these lamellar metal composites. And it contains the scope of the research and its originality. The reviewer was right that our topic was primarily about magnetism and not about absorbing properties. Considering the Reviewer’s suggestion, we have revised the expression appropriately in the revised manuscript.
- XRD in the fig. 3 didn’t explicitly revealed ferrites structures as the intensities of the respective reflections are almost invisible. SEM data didn’t contain any diffraction from the nanoparticles. The authors need to provide the evidence that the precursors were completely decomposed with the ferrites formation.
Reply: Thanks for the mention of the reviewer. the XRD pattern in figure 3 is revised on page 3, line 93.
- It’s unclear if magnetic properties of the FeSiAl flakes were affected after the heat-treatment. Please provide the respective measurements.
Reply: Thanks for the mention of the reviewer. For pure FeSiAl powder, the main factors that can affect its magnetic properties are composition and structural phase transformation. The magnetic properties of the DO3 phase of FeSiAl flakes were barely affected after the heat-treatment. In our study, the influence of heat treatment on the structural phase of surface coating is mainly discussed.
- No XRD data is provided for the heat-treatments at 650, 700, 750 and 850°C.
Reply: Thanks for the mention of the reviewer. The XRD data of CuFe2O4 coated flaky FeSiAl for the heat-treatments at 650, 700, 750, 800 and 850°C was provided in the supplementary information file.
5.Page 1, line 30: «absorbent» should be «absorber»
Reply: As reviewer suggested, we have revised it in the revised draft.
- Fig. 3: description for (a) and (b) sub-figures are missing
Reply: Thanks for the mention of the reviewer. In order to show the phase more clearly, we modified Fig. 3.
- Fig. 4 and 5: obviously (a), (d), (h), (k) and (n) sub-figures are too similar with each other. One sub-figure is enough. Others can be transferred into a supplementary information file.
Reply: Thanks for the mention of the reviewer. Compared to the pure FeSiAl powder as shown in Fig2a, the low-power images of Fig 4 and 5 (a), (d), (h), (k) and (n) sub-figures confirmed that the nano-ferrites are uniformly coated on the flaky FeSiAl surface rather than simply mixed, albeit with different ferrites.

Reviewer 4 Report
The title is informative and the abstract clear and well written, summarizing the results as well as what, why, and how the work was realized.
The entire work is well organized and clearly written, but is too concise having the aspect of an extended abstract. However I do believe but before publication in a prestigious journal as Nanomaterials is a need to be revised accordingly
1.The introduction is not able to sustain the paper novelty . This issue coul be a part of manuscript aim
2 A total of 21 references I do not think that is suitable for a paper in our time with a great papers number in this field
3. The paper topic is about morphology evolution and magnetic properties at different anneling temperature but the microstructural features modifications are not enough visible, labeled with arrows and consistent with the text descriptions of them
4. XRD peaks are not evidenced in figure 3
5 The table 1 it will be better to introduce some statistical treatment of data
Author Response
Reviewer 4
The title is informative and the abstract clear and well written, summarizing the results as well as what, why, and how the work was realized. The entire work is well organized and clearly written, but is too concise having the aspect of an extended abstract. However I do believe but before publication in a prestigious journal as Nanomaterials is a need to be revised accordingly
1.The introduction is not able to sustain the paper novelty. This issue could be a part of manuscript aim
Reply: Thanks for the mention of the reviewer. The introduction is revised and improved, and the research objectives are highlighted.
- A total of 21 references I do not think that is suitable for a paper in our time with a great papers number in this field
Reply: Thanks for the mention of the reviewer. We try our best to cite the most recent and directly relevant research findings in the field.
- The paper topic is about morphology evolution and magnetic properties at different anneling temperature but the microstructural features modifications are not enough visible, labeled with arrows and consistent with the text descriptions of them
Reply: Thanks for the mention of the reviewer. In fact, at the annealing temperature of 800°C, the differences of ferrite nanostructures are great as shown in Fig4 (c), (f), (j), (m) , and (p). In the case of cobalt ferrite, the microstructural features modifications are not enough visible with the increase of annealing temperature, but the structure will change abruptly at 850°C.
- XRD peaks are not evidenced in figure 3
Reply: As reviewer suggested, the XRD pattern in figure 3 is improved on page 3, line 93. But the diffraction peak of nano-ferrite is easy to be submerged under the strong diffraction peak of FeSiAl powder.
- The table 1 it will be better to introduce some statistical treatment of data
Reply: Following the reviewer’s comment, we have deleted the table 1 so as not to duplicate it. In fact, the treatment of data is already shown in Fig.6d.

Round 2
Reviewer 3 Report
Dear Authors,
Thank you very much for your answers. I think that you still need to add up more relevant references on magnetism and discuss them in Introduction section. That's why I recommend a minor revision.
Author Response
Reviewer 3
Thank you very much for your answers. I think that you still need to add up more relevant references on magnetism and discuss them in Introduction section. That's why I recommend a minor revision.
Reply: Thanks for the mention of the reviewer. We have added more five references on magnetism and discuss them.
Reviewer 4 Report
In my opinion despite the fact that authors operated some improvements such as a better subchapter conclusions, but the changes are not enough and the paper needs another revision. based on a detail support comparison of the content with other papers. Authors afirmed that 21 references are relevant and may be is true, but it is necessarly to present more clearly the original character
Author Response
Reviewer 4
In my opinion despite the fact that authors operated some improvements such as a better subchapter conclusions, but the changes are not enough and the paper needs another revision. based on a detail support comparison of the content with other papers. Authors afirmed that 21 references are relevant and may be is true, but it is necessarly to present more clearly the original character.
Reply: Thanks for the mention of the reviewer. Considering the Reviewer’s suggestion, we further have modified the conclusions. In addition, the references is true and we have updated them.